# Interventions to Reduce Implicit Bias in High-Stakes Professional Judgements: A Systematic Review

**DOI:** 10.3390/bs15111592

**Published:** 2025-11-20

**Authors:** Isabela Merla, Fiona Gabbert, Adrian J. Scott

**Affiliations:** School of Mind, Body and Society, Goldsmiths, University of London, London SE14 6NW, UK; f.gabbert@gold.ac.uk (F.G.); a.scott@gold.ac.uk (A.J.S.)

**Keywords:** implicit bias, decision-making, bias reduction, legal judgement, professional judgement, evidence-based practice

## Abstract

A systematic review was conducted to examine interventions designed to reduce the influence of implicit bias on professional judgements, with the aim of identifying strategies relevant to forensic and legal contexts. These decisions are often made under time pressure, ambiguity, and limited information, increasing reliance on intuitive judgement and mental shortcuts that can allow bias to shape how information is evaluated. Eight databases were searched and screened using predefined inclusion criteria. Studies were included if they assessed the behavioural impact of a bias-reduction intervention on decisions made by professionals or mock professionals in forensic, legal, healthcare, educational, or organisational settings. Thirty-eight studies met the inclusion criteria and were analysed. Interventions were mapped by mechanism, delivery format, and decision context. Systemic strategies, such as decision protocols, standardised rubrics, or changes to how information was presented, consistently outperformed individual-level approaches focused on changing attitudes or awareness. Effective interventions typically constrained discretion or embedded structured prompts at the point of judgement. However, most were tested in simulated settings, with limited evidence of long-term or applied effects. The review identifies strategies with the strongest empirical support and highlights those most effective, practical, and transferable to forensic and legal contexts.

## 1. Introduction

### 1.1. Implicit Bias Within the Criminal Justice System

Implicit bias refers to automatic associations and stereotypes based on social characteristics such as race, gender and age, which can influence judgements and behaviour without conscious intent. It is a form of cognitive bias that arises specifically in social contexts, where cues related to social characteristics can affect how people evaluate or respond to others, even when they are motivated to make impartial decisions ([24]). While the concept has been subject to debate and definitional variation ([11]; [23]; [29]), there is broad agreement that implicit bias can shape judgement and behaviour in ways that lead to discriminatory outcomes ([2]; [25]). This influence has been demonstrated across applied professional settings, including healthcare, education, employment, and particularly the forensic and legal context. This systematic review focuses on this latter topic, where the impact of bias on high-stakes decisions raises serious concerns about fairness and equity. Decision-makers in the forensic and legal context may be especially susceptible to the effects of implicit bias because decisions are frequently made under conditions of time pressure, ambiguity, and limited information. These conditions often allow for considerable discretion and foster reliance on intuitive judgement and mental shortcuts, which increase the likelihood that automatic associations based on social characteristics will shape how information is processed and evaluated ([8]; [7]; [16]).

Implicit bias has been shown to influence decision-making across multiple stages of the forensic and legal process, and among a wide range of professionals. For example, in the context of race, judges have been found to deliver harsher sentences to Black defendants than to White defendants, even when controlling for relevant legal variables ([43]). More recently, a Ministry of Justice analysis of 21,000 indictable Crown Court cases found that, compared with White defendants, Black, Asian, and Chinese had approximately 53%, 55%, and 81% higher odds of imprisonment overall, and within drug offences ethnic minority defendants had around 240% higher odds of receiving immediate custody ([30]). Prosecutors are also more likely to charge Black individuals with offences that carry mandatory minimum sentences, which can lead to longer periods of incarceration and fewer opportunities for plea negotiations ([52]). Pretrial decisions reflect similar disparities, with Black defendants more likely to be assigned higher bail amounts or denied bail, which increases the likelihood of guilty pleas and results in more severe sentencing ([58]; [13]).

Similarly, jurors may hold implicit beliefs about a defendant’s credibility, guilt, or dangerousness based on race or other social characteristics, which can subtly shape how evidence is interpreted. These biases often reinforce prior assumptions rather than promote objective assessment. In particular, jurors who implicitly associate Black defendants with criminality are more likely to interpret ambiguous evidence as proof of guilt and give disproportionate attention to the defendant based on race, even in the absence of overt racial content ([50]; [59]; [57]; [42]; [63]). Consistent with this, meta-analytic evidence shows small but reliable race-linked differences in mock-juror outcomes that typically reflect own-group favouritism. On average, White jurors are more punitive towards Black (and in some analyses, Hispanic) defendants, and Black jurors are more punitive towards White defendants ([42]).

Even outside the courtroom, implicit bias can affect how defence attorneys and death penalty lawyers advise clients, sometimes unconsciously adjusting their approach based on the client’s race ([17]). Forensic experts may also be influenced by racial cues, particularly in subjective or ambiguous cases, resulting in biased interpretation, presentation, or prioritisation of evidence ([15]).

### 1.2. Limitations of Current Intervention Approaches

Despite the serious consequences of implicit bias in forensic and legal contexts, very few interventions designed to mitigate its influence have been empirically tested, and most remain theoretically informed ([54], [55]; [35]). This is not only a gap in the criminal justice literature but also reflects a broader challenge across applied professional settings. In healthcare, for example, implicit bias contributes to disparities in diagnosis and treatment ([19]). Like forensic and legal contexts, it involves complex decision-making under time pressure and uncertainty; conditions that foster reliance on mental shortcuts and automatic associations. While numerous interventions have been trialled, most have shown little reliable impact on clinical decisions or behaviour ([61]).

A major reason for this lack of reliable impact lies in how existing implicit bias interventions have been designed and evaluated. Most established approaches focus on changing people’s internal attitudes, associations, or awareness, and are typically evaluated based on whether they produce changes on psychological tests, most notably the Implicit Association Test (IAT), which measures the strength of automatic associations between social categories and evaluative traits based on reaction times ([20]; [48]). In much of the literature, reductions in IAT scores are taken as evidence that an intervention has been successful. Yet IAT scores are poor predictors of real-world behaviour, and improvements rarely translate into better decision-making or fairer outcomes ([21]; [37]; [36]; [46]; [6]). Moreover, most studies are conducted in laboratory settings, offering little insight into how interventions function in professional contexts ([25]).

This raises concerns about the practical utility of research to date. Applied settings, including the forensic and legal context, require strategies that reduce biased outcomes by improving decision quality and consistency, and by limiting the influence of implicit bias on professional judgement. Yet, most existing research provides limited guidance on how to achieve these outcomes in practice. A recent meta-analysis of methods for reducing prejudice highlights this gap, concluding that despite the wide range of strategies tested, there is still little reliable evidence identifying which interventions lead to meaningful behavioural change ([48]). Without stronger evidence, it remains unclear which interventions should be prioritised for implementation in policy or adopted in professional contexts.

### 1.3. The Present Review

To address this gap, the present review synthesises empirical studies that have tested interventions designed to reduce the influence of implicit bias on consequential professional judgements. It focuses on decision-making contexts in which social characteristics such as race, gender and age are visible or can be inferred, and may shape evaluations in ways that create disparities. The review is designed to inform practice in forensic and legal contexts, and draws on research from fields where similar decision structures, challenges, and stakes are present to identify effective strategies in high-stakes environments and assess their potential for adaptation to legal and justice contexts.

To this end, the review will map the intervention landscape by identifying and describing the full range of strategies, specifying what each strategy changes, how it is delivered, and at what point in the decision process it operates. It will then evaluate the effectiveness of strategies by synthesising changes in decision outputs, focusing on the direction of effects, any impact on decision quality, and the durability of these effects where follow-up data are available. Finally, it will appraise the practicality and transferability of strategies by assessing whether interventions are feasible to implement within their original domain and whether their underlying mechanisms are likely to transfer to forensic and legal contexts, given their resource requirements, delivery conditions, time and operational constraints.

Three questions will be addressed. First, which intervention strategies have been tested to mitigate the influence of implicit bias in settings where evaluators make consequential decisions about other people? Second, how effective are these strategies when assessed against behavioural or evaluative outcomes, such as verdicts, treatment decisions, grades, shortlists, or hires? Third, which of these strategies can be adopted, scaled, and transferred to forensic and legal contexts, and under what conditions?

This review adds value in three ways. First, it focuses on behavioural and evaluative outcomes (i.e., what professionals do, decide, or recommend) rather than on changes in attitudes, such as shifts in belief, awareness, or self-reported bias, or changes in scores on bias-related measures. While attitude change has often been used as a proxy for progress, there is now substantial evidence that changes in implicit or explicit attitudes may not reliably translate into changes in behaviour, and that behaviour can change in meaningful ways even when attitudes remain unchanged ([2]). By assessing interventions in terms of whether they directly reduce disparities in decisions, actions, and practices, this review reflects a broader shift in the field toward evaluating real-world impact. It contributes to a more applied and outcome-focused body of evidence, and supports the development of strategies that respond more directly to the practical demands of professional decision-making.

Second, this review links effectiveness to specific contexts in which interventions are implemented, recognising that they need to work outside of experimental or highly controlled environments. Real-world conditions often involve time pressure, limited or incomplete information, and other potential organisational constraints. Thus, by situating effectiveness within context, this review helps identify interventions that are feasible in practice, as well as offering important insights into the resources necessary to support their implementation.

Third, the review distinguishes between strategies that show potential for adaptation and transfer across professional domains and those whose effectiveness appears confined to specific settings, thus highlighting the broader question of transferability. In this conduct, the review supports decision-makers and practitioners in selecting effective strategies and understanding their scope and limitations, particularly within forensic and legal contexts. To our knowledge, it is the first review to systematically draw on evidence from multiple professional domains to examine and evaluate its applicability to forensic and legal contexts.

## 2. Method

A systematic review was conducted to identify effective intervention strategies that have been empirically tested to reduce the influence of implicit bias in professional settings where evaluators make consequential decisions about other people. This review was pre-registered on PROSPERO (CRD420251107033) and followed a pre-specified protocol developed in line with guidance for systematic evidence synthesis ([47]).

The steps for constructing the systematic review involved a comprehensive search strategy, a structured screening process, data extraction, and a quality assessment of all included studies. Each of these stages is described below.

### 2.1. Search Strategy

In March 2025, a systematic search was conducted across eight electronic databases to identify eligible, published empirical articles. The databases were: APA PsycInfo, APA PsycArticles, Criminology Collection, ERIC, Social Science Database, ASSIA (Applied Social Sciences Index and Abstracts), PubMed, and Web of Science. These sources were selected to reflect the interdisciplinary scope of the review, covering psychology, social science, criminal justice, health, and education.

The search strategy combined terms across three conceptual domains: implicit bias, intervention, and decision-making. For bias, search terms included: implicit, unconscious, subconscious, automatic, heuristic, myth*, bias*, prejudic*, stereotyp*, attitude*, association*, discriminat*, and preference*. For interventions, terms included: debias*, intervention*, reduc*, training, strategy, method, program*, approach*, chang*, tool*, educat*, modif*, diminish*, counteract*, mitigat*, refram*, and effective*. Decision-making terms included: decision-making, decision task, judg*, evaluat*, deliberation*, and verdict*. These terms were combined into search strings and adapted to the search architecture of each database using Boolean operators and truncation symbols to maximise sensitivity. The full PsycInfo search string is provided in Appendix B.

Searches of titles, abstracts and keywords were conducted and were filtered to include only peer-reviewed journal articles published in English from 1 January 2000 onwards. This time frame was selected to reflect a documented shift in how bias and discrimination were conceptualised, including increased use of the terms ‘implicit bias’ and ‘unconscious bias’ in both scientific and public discourse ([25]). Studies from all geographical regions were considered, provided they met the inclusion criteria. The search was limited to studies involving adult human participants, and where possible, preprints, dissertations, and non-scholarly outputs were excluded to ensure that the knowledge obtained was peer reviewed. Full database-specific search strings, fields and filters are presented as a Appendix A, reproducing the exact configurations used across all eight databases. Beyond the database searches, to ensure comprehensive coverage, additional studies were identified through backward and forward citation tracking, and review of reference lists from previous systematic reviews on related topics.

### 2.2. Screening Process: Inclusion and Exclusion Criteria

Studies were included if they met four primary criteria:(1)Focus on implicit bias: The study targeted bias in social evaluations related to social characteristics (e.g., race, gender, age). Bias was considered ‘implicit’ if it was described as automatic, unintentional, or unconscious, including related constructs using terms such as ‘automatic stereotyping’ or ‘unconscious associations’. Implicit bias was operationalised as a socially focused subtype of cognitive bias (i.e., a tendency for evaluative judgements to be influenced by automatic, unintentional processing of social characteristics). Accordingly, studies centred on non-social cognitive heuristics (e.g., anchoring, confirmation) or on explicit cultural attitudes/beliefs not characterised as automatic processes were ineligible and were excluded. Eligibility was therefore evidenced either by the authors’ framing of bias as automatic/unconscious social processing or by an empirical design that operationalised bias as a change in decisions produced by a social characteristic in the absence of task relevance or explicit intent (e.g., an evaluative decision task that manipulated a social characteristic while non-task-relevant to assess its effect on decisions).(2)Tested an intervention: The study examined an intentional strategy aimed at reducing or mitigating implicit bias in decision-making. Studies that reported incidental bias reduction without presenting it as a deliberate intervention were excluded.(3)Decision-making outcome: Outcomes had to reflect a meaningful judgement with real or simulated consequences about another person, such as sentencing, hiring, grading, performance evaluation, or treatment recommendation. Studies focused on attitudes, preferences, or affective ratings without an evaluative consequence were excluded.(4)Professional or mock-professional context: This included both real professionals acting in their formal roles (e.g., doctors, teachers, police officers) and lay participants who were explicitly instructed to take on a professional role (e.g., mock jurors, hiring decision-makers). Studies where outcomes could not be linked to an individual decision-maker were excluded.

In addition to the above, studies were excluded if they were non-empirical (e.g., literature reviews or theoretical articles), not published in English, and unpublished, including dissertations and preprints, to ensure peer-reviewed quality.

### 2.3. Screening Process: Title, Abstract, and Full Text Review

All articles retrieved through the database searches were exported into Excel, where duplicates were identified and removed. Titles and abstracts of the remaining articles were screened for relevance against the inclusion criteria. Full texts were then retrieved for all articles that appeared to meet the inclusion criteria or for which eligibility was uncertain based on the abstract.

Screening was conducted in two phases. In the first phase, the first author screened the titles and abstracts of all identified articles. If the first author was unsure about the inclusion of any studies, this was discussed further with the Research Team. In the second stage, the full texts of potentially eligible articles were retrieved and assessed against the full inclusion criteria by the first author and checked by the second author. Discrepancies were resolved through discussion and reference to the protocol, with input from the third author where necessary. Reasons for exclusion at the full-text stage were recorded.

The screening process adhered to PRISMA guidelines. Figure 1 presents a PRISMA flow diagram that summarises the number of records identified, screened, assessed for eligibility, and included in the final review. In total, 38 studies from 26 articles met the inclusion criteria and were included in this review.

### 2.4. Data Extraction

A Searchable Systematic Map (SSM) was developed in Excel to represent the 38 studies that met the review’s inclusion criteria. Each study was systematically coded using a structured framework designed to support cross-study comparison, analysis, and inform the narrative synthesis. The SSM functioned as both a descriptive summary and an analytical tool, allowing the evidence to be filtered, examined, and compared across key dimensions to explore what worked, how it worked, and how likely it was to be transferable to applied professional settings.

Data extraction was guided by four domains: 1. Study details, 2. Intervention characteristics, 3. Outcomes and findings, and 4. Delivery practicality and implementation feasibility. Full definitions and coding rules are provided in Appendix C.

(1)Study details: Captured core methodological and contextual information (e.g., year, country, professional field, study design, participant role, paradigm). Three ecological indicators (sample realism, task realism, and context realism) were also coded to assess the ecological validity of each study and the extent to which findings might generalise to applied professional settings.(2)Intervention characteristics: Documented each intervention’s design, delivery, mechanism of action, and timing. Interventions were classified both by level of operation (individual, systemic, or mixed) and by mechanism (e.g., altering information available at judgement, adding structure to reduce discretion, prompting self-regulation, reframing assumptions, or targeting automatic associations). These classifications facilitated structured comparison of strategies and their potential transferability.(3)Outcomes and findings: Extracted evidence on whether interventions reduced bias in consequential decisions (e.g., sentencing, hiring, grading, treatment). Effectiveness was judged against baseline bias and assessed for statistical significance, consistency, and durability. Secondary outcomes (e.g., implicit bias measures, participant feedback) were also recorded to provide contextual insight into mechanisms and broader impact.(4)Delivery practicality and implementation feasibility: Recorded information on format, materials, time demands, training needs, and scalability. Interventions were appraised for practicality (resource and process requirements) and feasibility (likelihood of adoption, fidelity, and sustainability under real-world constraints). Ratings of high, moderate, or low were applied across key aspects such as cost, facilitation needs, duration, and scalability.

Together, these domains allowed the review to assess not only whether interventions were effective, but also how they operated and how feasible they would be to implement in professional practice, particularly in high-stakes forensic, legal and related contexts. For more information about the data extraction process, please see Appendix C.

### 2.5. Quality Assessment

A structured critical appraisal was conducted to assess the internal validity, methodological quality, and reporting transparency of all included studies. The purpose was to strengthen the reliability of the synthesis by identifying the strength of evidence behind each reported effect and to avoid over-weighting findings from studies with design limitations. Appraisal was not used to exclude studies unless fundamental flaws affected meaningful interpretation.

Appraisal followed the Quality Assessment Tool for Diverse Studies (QuADS), a published tool developed for systematic reviews, selected for its applicability across diverse empirical designs ([27]). The tool covers 13 domains, including clarity of theoretical framework, appropriateness of study design and sampling strategy, and transparency in data collection, recruitment, analysis methods and stakeholder involvement. Each domain was scored from 0 (not at all) to 3 (fully), based on information reported in the publication. Following QuADS guidance, domain scores were interpreted qualitatively and synthesised into an overall judgement of low concern (most domains scored 2–3), some concerns (mixed scoring patterns with one or more domains scored at 1), or high concern (multiple domains scored 0–1), regarding methodological and reporting adequacy, based on predefined criteria. Full domain descriptions and scoring anchors are provided in Appendix D.

## 3. Results

### 3.1. Overview of Study Characteristics

The review includes 26 articles reporting 38 distinct studies. Studies were published between 2002 and 2024, with the majority (73.7%) published from 2020 onwards and half (50.0%) between 2021 and 2024. Earlier studies were infrequent, with a smaller number of articles published between 2002 and 2019 (26.3%), reflecting a recent increase in applied debiasing research. Most studies were conducted in the United States (63.2%), followed by Western Europe (26.3%) and Asia (10.5%).

Most studies investigated bias in workplace or organisational settings (63.2%), followed by criminal justice (13.2%), healthcare (13.2%), education (7.9%), and civic or political decision-making (2.6%). The most targeted forms of implicit bias were race/ethnicity (42.1%) and gender/sex (57.9%). Age was included in fewer studies (10.5%), followed by socioeconomic status (7.9%). A small subset of studies (15.8%) investigated more than one social characteristic within the same task or across separate comparisons. Interventions that were investigated were designed to operate at the individual-level (47.4%) and the systemic-level (47.4%) in equal measure, with a small number combining both (5.3%). Online delivery was more common (63.2%) than in-person delivery (36.8%).

Ecological realism was mixed. Sample realism was high in around two-fifths of studies (39.5%), where participants were practising professionals or professionally relevant trainees. It was low in a considerable proportion (36.8%), typically involving general student populations or online panels with limited applied relevance, and moderate in a smaller number (23.7%), which involved working adults or students with some professional relevance. Task realism was predominantly moderate (55.3%), characterised by simplified but recognisable approximations of real decision scenarios. It was low in some cases (34.2%), with abstract or overly artificial formats, and high in a few studies (10.5%) where tasks closely reflected the complexity and structure of high-stakes decision-making. When considering sample, task, and paradigm together, overall context realism was low in half of the studies (50.0%), followed by moderate (39.5%), and high in only a small subset (10.5%). In practical terms, most of the interventions were tested on simplified versions of professional decisions within lower-stakes, survey-based environments.

The timing of intervention delivery was typically close to the decision point. Interventions were delivered immediately before the decision in half of the studies (50.0%) and during the decision process in a smaller subset (44.7%). Only a small number (5.3%) introduced the intervention earlier, at a greater temporal distance from the judgement. Durability was rarely assessed. Only a minority of studies (15.8%) included any follow-up measurement, with time intervals ranging from one week to approximately ten months. The remainder (84.2%) measured outcomes only immediately after the intervention. Studies providing conclusions about the persistence of intervention effects were therefore limited.

### 3.2. Overview of Intervention Effectiveness

The findings show a clear pattern of intervention effectiveness by intervention approach. Systemic-level strategies, which target the decision environment, accounted for most strong effects—whereby effects were labelled as strong when an intervention produced a clear, sample-wide main-effect reduction on the targeted decision outcome, moderate when improvements were conditional (e.g., limited to subgroups, specific measures or interaction-only patterns) rather than uniform across the full sample, and limited when there was no effect on the targeted outcome or when the study lacked a baseline disparity on that outcome. Of the eighteen systemic-level studies identified, fourteen produced strong effects (77.8%). Contrastingly, individual-level strategies, which target the decision-maker, showed strong effects in seven of eighteen studies (38.9%), while mixed-level interventions, combining changes to individual decision-making with structural adjustments, produced mostly conditional or limited evidence. These patterns are examined in detail below.

### 3.3. Systemic-Level Interventions

Systemic strategies produced the most consistent improvements, accounting for two-thirds of all strong findings in this review (66.7%). These interventions were organised into two recurrent mechanisms: 1. Altering information available at judgement; and 2. Adding structure that limits discretion.

#### 3.3.1. Altering Information Available at Judgement

Ten studies examined interventions that changed what decision-makers saw at the decision point or how this information was presented (26.3% of all studies; 55.6% of systemic studies). Seven studies within workplace contexts examined gender (71.4%) or ethnicity (28.6%) in shortlisting or hiring tasks ([18]). Two studies targeted sex/gender and race/ethnicity in professional screening ([22]; [49]), and one examined gender bias in political committee selection ([62]). Overall, seven of ten interventions were effective (70.0%).

Changing what candidates were shown at the decision point was the most consistently effective strategy. All seven shortlisting and hiring studies produced more balanced shortlists and, when a single choice was required, increased the likelihood that a woman or minority candidate was selected ([18]). Partitioning candidates by social category encouraged selectors to distribute their selections across categories and outperformed brief prompts that only stated category information, indicating that the display design, rather than information alone, drove the effect. Grouping also altered selections. In these tasks, profiles from one category were shown together while profiles from the comparison category remained listed individually. This layout shifted attention to the individually listed profiles. Grouping the majority increased the selection of minority candidates, whereas grouping the minority reduced it. Where assessed, these gains did not reduce decision quality, though stronger implicit biases (higher gender IAT scores) weakened effects. By contrast, interventions that modified identity cues to prevent stereotype activation or added process transparency to encourage more proportionate exploration produced limited results. In STEM hiring, standardising an availability cue by adding the same ‘long-hours’ note to matched male and female CVs, in the form of a generic statement of willingness to work extended hours, shifted how managers weighed that criterion but not hiring likelihood ([22]). In residency screening and political committee formation, there was no measurable change from controls ([49]; [62]).

These studies offered limited evidence for real-world impact. Most evaluations used non-professional samples and simplified tasks. Only one shortlisting study used an in-person paper-resume procedure and reproduced the display effect ([18]). Residency redaction and STEM screening involved practising faculty or managers, yet outcomes did not differ between the compared groups, leaving no baseline disparity against which to assess change. None of the studies included follow-up measurement; however, for systemic strategies, the central issue is not retention, but whether protocols are implemented with high fidelity and without unintended consequences such as accuracy errors. Future replications with professional decision-makers should include fidelity checks and assess decision quality and error rates, alongside equity outcomes, to establish suitability for high-stakes practice.

#### 3.3.2. Adding Structure That Limits Discretion

Eight studies tested interventions that constrained bias-prone discretion by adding structure to decision processes (21.1% of all studies; 44.4% of systemic studies). Six studies focused on workplace hiring or shortlisting (75.0%) ([3]; [60]; [40]), one on education grading (12.5%) ([51]), and one on clinical care management (12.5%) ([26]). Most interventions targeted gender (75.0%), with race/ethnicity considered in two studies (25.0%). Seven out of eight interventions were effective (87.5%).

Interventions that required decision-makers to follow standardised procedures or predefined evaluative criteria produced the most consistent improvements in outcomes. In healthcare, a structured labour-management protocol reduced caesarean rates and improved neonatal health for Black patients, without adverse effects for others ([26]). In education, grading with a predefined, detailed rubric eliminated racial disparities that were otherwise observed under more subjective conditions ([51]).

Hiring studies applied comparable strategies to constrain discretion in candidate evaluations. Here, structured interviews with behaviourally anchored rating scales reduced subjectivity and minimised stereotype-driven assessments ([3]). Another intervention asked evaluators to assign weights to selection criteria before reviewing applications. This pre-commitment eliminated the gender preference typically shown by male raters ([60]). Three other hiring studies increased shortlist length by requiring evaluators to identify six candidates rather than three, which increased the number of women but found no corresponding change in final selections ([40]).

These studies offer strong evidence that standardised protocols and predefined criteria can improve decision-making by limiting the role of discretionary judgement. Most of the hiring studies using this mechanism showed similar promise but often relied on student or online samples using simplified tasks, leaving generalisability to high-stakes hiring decisions unclear. Similar to the previous mechanism of systemic interventions, none of the studies assessed outcomes beyond the immediate decision or evaluated decision quality. Conclusions about fidelity use and error rates are, therefore, limited and should be investigated by future research.

### 3.4. Individual-Level Interventions

Individual-level strategies contributed to one-third of all strong findings in this review (33.3%). These interventions were organised into three categories: 1. Prompting self-regulation at the point of decision, 2. Reframing assumptions, and 3. Targeting automatic associations.

#### 3.4.1. Prompting Self-Regulation at the Point of Decision

Nine studies encouraged individuals to pause, reflect, or engage in corrective routines before making a judgement (23.7% of all studies; 50.0% of all individual studies). Most were conducted in workplace evaluation or selection contexts (44.4%) ([1]; [14]; [34]), followed by criminal justice decision-making (44.4%) ([41]; [53]; [31]), and school disciplinary decisions (11.1%) ([44]). Targeted biases included race/ethnicity (55.6%), gender (44.4%), age (33.3%), and socioeconomic status (11.1%). Three of these studies were effective (33.3%).

The clearest evidence for real-world impact came from a field-based policing study that combined classroom instruction with high-fidelity simulation training. Officers learned about how bias can influence judgement, concrete practices for fairness and de-escalation, and then practised in simulators with immediate feedback. This study produced sustained improved performance in real-world interactions, particularly with community members low in socioeconomic status, and reduced discrimination-related complaints over the 10-month period of monitoring ([31]).

Consistent improvements also came from brief prompts embedded directly into the decision workflow, designed to encourage reflection and direct attention toward relevant evaluation criteria. In workplace hiring, an on-screen reminder that age can bias evaluation shifted attention to job-relevant skills, which narrowed or eliminated age gaps in ratings without reducing attention to applicant qualifications ([34]). A similar intervention in education added a ‘pause and plan’ step to school discipline procedures. This lowered teachers’ referral rates for Black students and shifted perceptions of those students’ behaviour, suggesting a move toward more deliberate judgement ([44]). Some improvements were also observed in two HR studies, where short age-bias warnings eliminated penalties against older applicants in both performance appraisal and hiring tasks. However, accountability requirements helped older men more than older women, pointing to selective effects across intersecting identities ([14]).

By contrast, prompts lacking clear behavioural guidance or when baseline disparities were absent showed weaker results. Two jury studies tested implicit-bias instructions and orientation videos, finding these increased bias-related discussion during deliberation, but not verdict disparities, and sometimes backfired and encouraged overcorrection ([41]; [53]). Similarly, in workplace leader evaluations, prompts produced inconsistent results that were moderated by raters’ implicit attitudes ([1]).

Ecological realism varied widely, with most studies relying on simplified decision tasks, limiting what can be inferred about generalisability to high-stakes settings. An exception was that of [31] ([31]), which embedded training into professional routines and demonstrated that self-regulation works best when paired with clear behavioural strategies and feedback. Where individual differences were tested, effects were uneven, raising questions about scalability and equity across intersectional identities.

#### 3.4.2. Reframing Assumptions

Five studies tested interventions that reshaped how decision-makers construe people and attribute capabilities (13.2% of all studies; 27.8% of individual studies). Three were conducted in workplace evaluation or selection contexts (60.0%) ([12]; [39]), followed by two in healthcare treatment decisions (40.0%) ([28]; [45]). Targeted biases included race/ethnicity (40.0%), gender (40.0%), age (20.0%) and socioeconomic status (20.0%). Three interventions were effective (60.0%).

Across studies, the most consistent effects came from interventions that explicitly surfaced bias in decision-makers’ own judgements and redirected them toward individuated, evidence-based appraisals rather than group-based assumptions. In healthcare, one intervention delivered real-time feedback on treatment decisions with virtual perspective-taking modules, increasing empathy and equity in pain management ([28]). A separate intervention reframed bias as a shared responsibility and paired it with guidance on shared decision-making, reducing age-based assumptions in cancer care ([45]).

In a workplace evaluation, one study tested two training formats aimed at promoting more individualised judgement of Arab/Moroccan applicants. One showed realistic workplace misunderstandings between majority and minority employees, after which participants made judgements, received corrective feedback, and then engaged in a debrief session with discussion and role-play. This approach built more nuanced mental models of cultural difference and improved job suitability ratings. The second task asked participants to study scripted workplace vignettes, then recall specific behaviours into key performance domains before rating the target. This also improved ratings, though effects were smaller. For both, improvements faded by three months, highlighting the need for reinforcement ([12]). By contrast, two studies used short reframing prompts to shift assumptions about who is seen as a ‘leader’. Both introduced a universal framing, emphasising that leadership potential is widespread and developable, and tested its effects on gender bias in candidate evaluations. Effects were mixed and, in one case, no bias was observed under control conditions, limiting conclusions ([39]).

Overall, healthcare interventions aligned most closely with real-world practice, while workplace studies often relied on online evaluations by students or adults with varied experience. Durability remained limited, with effects fading without reinforcement.

#### 3.4.3. Targeting Automatic Associations

Four studies attempted to directly alter implicit associations (10.0% of all studies; 22.2% of individual studies), three in workplace hiring or evaluation (75.0%) ([32], [33]; [4]), and one study in criminal justice (25.0%) ([56]). Bias targets were split between gender (50.0%) and race/ethnicity (50.0%). One intervention was effective (25.0%).

Within these studies, the consistent effect came from an intervention that reshaped automatic associations by increasing perceived variability within social groups, thereby reducing reliance on stereotypes at the point of decision. In a hiring simulation, participants prompted to complete simple variability sentences about Arab individuals (e.g., “Whereas some…, other…”) showed no bias in composite ratings, rankings, or interview selections, unlike participants exposed to homogeneity prompts or controls. Mediation analyses showed that the manipulation increased perceptions of within-group variability (i.e., diversity), and this increase accounted for the reduction in biased evaluations ([4]).

By contrast, counter-stereotype retraining did not reliably reduce gender bias in candidate selection, and sometimes overcorrected ([32], [33]). In criminal justice, a brief VR task that placed participants in a Black avatar reduced race-IAT scores and increased confidence in not-guilty verdicts. However, no baseline disparity was present in the control group, limiting what can be inferred about bias reduction ([56]).

Ecological realism was generally low: all four studies used laboratory-style simulations with student or community samples. Follow-ups were rare and inconclusive. Overall, targeting automatic associations shows proof of concept for altering representations, but evidence of durable impact in real-world settings is lacking.

### 3.5. Mixed-Level Interventions

Mixed strategies were rare, appearing in only two studies (5.6%). One addressed race in clinical interactions ([10]), the other socioeconomic status and ethnicity in educational evaluations ([38]).

Neither produced strong effects, but suggested promising avenues for integrating skills-based regulation with low-cost process prompts. The healthcare study combined brief pre-learning with high-fidelity simulation to train implicit bias mitigation strategies such as individuation, partnership building and perspective-taking, reinforced through immediate feedback and reflective debriefs. Modest improvements in judgement scores and some unit-level indicators, such as a decrease in security dispatch calls during and after the intervention, were observed immediately and at three-month follow-up. However, results were descriptive and lacked a control group. In the education study, a short online module combined theory-based instruction with two process constraints: slowing down and applying criteria, and an ‘if-then’ intention before each decision. Teacher judgements of low socioeconomic students improved, particularly for academic capability. However, other effects were small or inconsistent.

### 3.6. Practicality of Interventions

Across the 38 studies reviewed, only two interventions (5.3%) demonstrated strong effects on decision outcomes in applied professional settings. These differed both in intervention approach, with one systemic and one individual, and in their practicality, reflecting broader patterns whereby systemic interventions were generally easier to implement.

The first intervention involved a standardised protocol that reduced racial disparities amongst practising clinicians ([26]). The second was a police training intervention that improved real-world behaviour and reduced discrimination-related complaints over approximately 10 months ([31]). The clinical protocol required only modest investment for development and monitoring and no additional time at the point of care, making it relatively low in cost and high in duration practicality ([26]). In contrast, the policing programme required about 12 h of delivery time, access to simulation facilities, and trained instructors, resulting in low practicality for cost, facilitation and duration ([31]). Together, these two studies offer rare examples of strong impact in applied professional settings, suggesting that effective interventions exist across different delivery demands: protocols suit contexts with limited training time or facilitation capacity, whereas classroom learning and simulation-training programmes are more suitable for organisations able to invest in extended delivery time, trained facilitators, and access to specialised equipment.

Among effective studies conducted under lower ecological realism, systemic-level strategies were the most practical to deliver and implement. Here, thirteen effective systemic interventions were highly practical (100.0%) relying on simple adjustments to decision flows, such as grouping candidates in their display ([18]), extending shortlists ([40]), or predefining criteria and including structure ([3]; [51]). These could be delivered in seconds to minutes without facilitators, all fitted within existing workflows with negligible marginal cost, and none required specialist involvement for delivery.

Effective or promising individual-level approaches were also practical when they were brief and embedded directly in the workflow. Across seven effective individual strategies, four were also highly practical (57.1%). These included anti-bias or self-regulation prompts ([34]; [44]) and simple accountability or warning prompts ([14]), which were mostly self-guided and required no facilitation. In contrast, the remaining three interventions (42.9%) were less practical. These included intercultural or structured recall training ([12]) and a clinician module that combined real-time treatment feedback with perspective-taking videos ([28]). These interventions were longer and required either trained facilitators or dedicated delivery platforms. Overall, practicality declined as dependence on trainers, specialised environments, or lengthy formats increased.

### 3.7. Transferability of Interventions

The most effective interventions varied not only in delivery demands but also in how easily their strategies could be transferred across domains and professional settings. While practicality describes how feasible an intervention is to deliver in its original format, transferability reflects whether that intervention could plausibly work in other contexts or be adapted to different organisational structures. While many effective interventions have features that support transfer, only a minority were tested outside their original setting, and even fewer assessed whether effects held under different conditions or over time. Therefore, suggestions are made with caution.

Among the few studies showing strong real-world effects, only one clearly offered a transferable potential. The standardised clinical protocol introduced in obstetrics was integrated into routine care and operated through pre-specified thresholds and actions, making it adaptable to any context where decision points are standardised and outcomes are routinely monitored ([26]). The underlying mechanism, specifically removing discretion through rule-based structures, applies broadly wherever decisions follow a predictable structure or are repeated. In contrast, the police training programme that combined instruction with simulator exercises was effective but less transferable, as it relied on dedicated equipment and trained facilitators, limiting use to settings with similar resources ([31]). The core idea of skills-based practice with feedback could generalise, but only with substantial investment or simplification.

Transferability was also strong among effective systemic strategies tested under lower ecological realism. These relied on structural features common across decision-making contexts, such as shortlists, application forms, or performance rubrics, and worked by changing how information was presented or how choices were constrained ([18]; [40]; [60]). Because they are embedded in standard formats and require little facilitation, they could be easily utilised across settings and implemented through policy changes, interface updates, or template modifications. In principle, these mechanisms are deployable to all the contexts where evaluators compare or assess individuals under conditions of discretion and potential ambiguity.

In contrast, individual strategies showed more uneven potential for transfer. Prompts that are brief, self-guided, and delivered at the decision moment are broadly applicable across settings. For instance, a short screen reminder about age bias could be added to any hiring or appraisal interface with no structural modifications ([34]). Similarly, reflective cues and pause instructions fit naturally into decision workflows in education and HR and could transfer with only minor contextual tailoring ([44]; [14]). However, interventions that rely on facilitators, video debriefs, or customised platforms are less transferable because they require specific technical infrastructure, such as interactive feedback systems or high-fidelity simulations ([28]). In these cases, the mechanism of the interventions could still be transferable, but the format of the delivery would require major redesign.

### 3.8. Critical Appraisal

All included studies were critically appraised using the QuADS tool. Consistent with guidance, scores were used to interpret confidence in findings and to highlight where evidence was strongest or required caution, rather than to exclude studies or impose numerical weights. Domain-level QuADS scores were recorded for each study and used to identify common strengths and weaknesses across the evidence base. The ratings were not intended to compare or rank study quality, but to clarify where methodological or reporting issues were most evident and how these shaped confidence in overall conclusions. This structured approach ensured consistency across studies of differing designs and helped identify recurring methodological and reporting patterns. Table 1 presents a QuADS summary of domain ratings (mean, standard deviation and range), accompanied by brief interpretative summaries.

Across the 38 studies, overall reporting quality was sufficient to support synthesis. Importantly, no study raised high concern, while ten indicated strong reporting and design clarity ([12]; [22]; [26]; [28]; [31]; [38]; [41]; [51]; [53]). The remaining 28 presented appropriate designs and analyses, but raised some concern in at least one domain, most often sampling justification, stakeholder involvement, or procedural transparency. Therefore, these were interpreted with greater caution, specifically with regard to their generalisability to applied professional settings.

Sampling was the most recurrent weakness. Only eight studies (21.1%) provided detailed justification, aligning samples with the study aims and/or offering power arguments ([22]; [26]; [41]; [51]; [53]; [14]). Eleven studies (28.9%) offered partial justification, typically linking sample characteristics with aims ([10]; [12]; [28]; [31]; [38]; [39]; [44]; [49]), while the remaining 19 studies (48.5%) provided only minimal sampling information ([1]; [3]; [4]; [18]; [32]; [33]; [34]; [45]; [56]; [62]). Despite a focus on professional decision-making, many studies either did not prioritise or did not document sample realism, recruitment transparency, or power planning. In the narrative synthesis, these sampling limitations were used to weight interpretations toward studies with transparent, appropriate sampling and higher ecological realism.

Stakeholder engagement was also limited overall. No study showed considerable stakeholder involvement. Six studies (15.8%) showed moderate evidence, including piloting that informed design or involvement by domain experts ([10]; [22]; [26]; [28]; [31]; [49]). Fourteen (36.8%) evidenced minimal involvement, for example, limited piloting ([3]; [12]; [18]; [41]; [44]; [45]; [53]). The remaining 18 studies (47.4%) reported no stakeholder involvement ([1]; [4]; [32], [33]; [34]; [38]; [39]; [51]; [56]; [62]; [14]). Therefore, evidence on feasibility, acceptability, and fit with existing professional workflows was sparse.

Measurement choices were typically sensible for isolating social-characteristic cues, but many tasks and instruments were author-designed rather than validated scales or established scenarios. This is not inherently problematic, as custom materials allowed several studies to hold task content constant across conditions (e.g., [1]; [12]; [51]). However, this introduced heterogeneity in how implicit bias was operationalised and detected, and psychometric properties (e.g., reliability, validity, invariance) were rarely reported. In addition, a small subset of seven studies (18.4%) reported absent or atypical baseline disparities in their primary decision outcomes, limiting any meaningful conclusion about bias reduction ([39]; [41]; [49]; [53]; [56]; [62]). Although a lack of baseline bias is not, in itself, a threat to internal rigour, it does suggest that the stimuli or outcome measures may have lacked diagnostic sensitivity to elicit or detect bias.

## 4. Discussion

This review synthesised 38 studies that tested interventions intended to reduce the influence of implicit bias on consequential judgements. Across the evidence base, a consistent pattern emerged between intervention approaches, suggesting that changing the decision environment was more effective than trying to change decision-makers. Among systemic-level interventions, 14 were effective (77.8%), compared to only seven (38.9%) individual-level interventions. The two mixed studies yielded only conditional or limited evidence. Of the 21 studies that produced strong outcomes, two-thirds (66.7%) were achieved by systemic changes.

These results align with the growing critique around individual-level bias training. While earlier implicit bias interventions have often targeted implicit associations or awareness, there is now a broad consensus that these changes rarely translate into behavioural improvements in applied settings ([2]; [21]; [20]; [48]). This review supports this view and points to a more practical conclusion: in high-discretion fields like forensic and legal contexts, where time pressure, ambiguity and variability are common constrains ([9]; [16]), interventions should prioritise the design of the decision environment, with individual strategies used selectively to support key skills or processes.

Most of the systemic strategies worked through two mechanisms. The first was altering what information was visible at the point of judgement. Here, simple display changes, such as partitioning candidates or grouping options, shifted attention and changed which comparisons were made without impacting decision quality, while increasing diversity of choice for women and minority candidates ([18]). The second mechanism was adding structure that limits discretion through predefined criteria, standardised rubrics or protocols ([3]; [26]; [40]; [51]; [60]). These interventions prevented evaluators from unconsciously shifting standards based on identity cues and made decisions more consistent and less reliant on subjective interpretation. Taken together, the most effective strategies intervene directly at the point of decision to constrain the influence of social categories and direct judgement towards evidence-based criteria.

This has clear relevance for contexts where redacting identifying details is not always feasible or desirable, including forensic and legal contexts. Pre-committing to legally relevant criteria before seeing identifying information could reduce shifting standards in complex judicial decisions, whereas structured decision aids, such as behaviourally anchored rubrics, could support any decisions that involve multiple, observable case-linked elements (e.g., credibility assessments, sentencing recommendations). Therefore, systemic strategies could aid in legal or forensic judgements where multiple people or claims must be considered at once, through structuring the presentation of information to reduce salience asymmetries and encourage comparisons on case-relevant grounds rather than intuitive or identity-based cues. Importantly, these strategies do not require evaluators to change their beliefs or reasoning style, only to operate within a process that improves the consistency and traceability of judgement.

Where decision quality was evaluated, improvements did not appear to coincide with reductions in accuracy, though this was rarely assessed. Going forward, implementation should include monitoring for fidelity, error rates, and unintended effects. Systemic strategies rely on consistent use to be effective; however, they can be embedded in forms, protocols, and digital systems that allow close monitoring. This also makes them attractive from a governance perspective, offering tools not just for reducing bias, but for improving transparency and accountability.

The findings on individual-level interventions were more mixed. Where effective, these strategies tended to include both a corrective goal and an opportunity to practice decision skills under realistic or time-pressured conditions. This was best illustrated in a field study of police officers, where a training programme combining classroom instruction with simulation improved real-world behaviour and reduced complaints over a sustained period ([31]). Other promising examples included short prompts embedded into decisions, designed to direct attention to relevant criteria or encourage reflection, with effects observed on selection, appraisal, and disciplinary outcomes ([34]; [44]; [14]). Additionally, three studies ([12]; [28]; [45]) used structured reframing interventions to prompt evidence-based evaluation, increase empathy, or reduce reliance on group-based cues, with observed effects on hiring, healthcare, and treatment decisions. Thus, the effective individual-level strategies worked not by changing underlying attitudes, but by interrupting automatic judgement, increasing motivation to individuate and supporting more effortful, evidence-linked reasoning.

By contrast, most interventions focused solely on awareness or association retraining showed little impact. Awareness-only prompts that lacked behavioural guidance sometimes triggered overcorrection, as in jury studies where participants misinterpreted justified scepticism as bias ([41]). Some prompts produced uneven effects across intersecting identities, helping some subgroups but not others ([14]). In other cases, interventions worked only for individuals with low implicit bias scores, and reversed for those with higher scores, raising concerns about unintended consequences ([1]). These findings reflect the complexity of implicit bias as a construct and the difficulty of targeting it directly without clarity on how it operates in decision-making. This points to the broader limitation of individual-level interventions being rarely precision-targeted and often resting on assumptions about mechanisms (e.g., that association change leads to behaviour change) that may not hold in practice ([21]). These approaches can still have value, but they need to be used carefully and only where there is clarity on what they target and how that connects to better decisions. Nonetheless, these results reinforce the view that targeting implicit bias through attitude change alone is unlikely to produce robust decision improvements.

In terms of transferability, the pattern is also clear and favours systemic interventions. What embeds most easily into workflow is also likely to scale well across contexts and roles. As such, changes to forms, processes, and displays are both practical to implement, typically requiring no facilitation, minimal costs and time burden, and highly transferable, particularly where decisions are repeated and structure can be applied without disrupting professional judgement. However, these same conditions that make these approaches transferable in principle may also constrain their implementation in practice. Specifically, in high-stakes domains such as forensic and legal work, professionals most affected by bias-related outcomes often operate under heavy time pressures, limited resources, and strict accountability demands. These realities can restrict both the feasibility and willingness to adopt new interventions, particularly where their potential influence on decision outcomes remains uncertain or where practitioners may later be required to defend their methods in court. Recognising these constraints is essential for interpreting how intervention findings translate beyond experimental settings, and for understanding why even well-evidenced strategies may prove difficult to embed within systems already functioning under significant procedural and ethical demands. Therefore, as transferability is shaped by both intervention design and the conditions under which they must operate, interventions that align with existing workflows and accountability structures have the highest potential to be sustained once implemented.

By contrast, resource-intensive strategies, such as high-fidelity simulations, while effective, remain more difficult to scale. However, their underlying logic, such as skill rehearsal and feedback, could still be approximated through lower-cost options, including video-supported reflective practice or digitally delivered simulations; though, such adaptation would need careful piloting to ensure they still deliver the same effects. These differences in practicality and transferability matter for high-stakes environments like forensic and legal contexts. In such settings, even modest improvements in decision quality could have significant downstream effects. Thus, systemic strategies that can be more readily implemented into existing processes offer a scalable route to improving consistency and fairness.

## 5. Conclusions

To advance from promising ideas to applied solutions, more research needs to test interventions under realistic constraints. Most studies in this review used simplified tasks, and mock professionals or adults with no professional relevance. Ecological realism remained limited across much of the evidence base, with only two studies (5.3%) demonstrating strong effects in high-stakes applied settings ([26]; [31]). This limited what could be inferred about transfer to real-world contexts. To address this important gap, future applied testing should involve professional decision-makers, more realistic decision formats and tasks, stakeholder-informed designs and measurements that capture error, fidelity, and suitability for the intended setting. Systemic interventions, in particular, should be evaluated not just for effect but for how they are used and translated in practice, including whether criteria and protocols are closely followed, and how reliably decision outcomes align with the intended standards.

Nevertheless, the applied potential of systemic strategies remains high, particularly for professional settings where decisions follow a predictable structure and are time-pressured and subject to wide discretion, such as forensic and legal contexts. The results reflect a broader shift in the field, after years of focus on reducing IAT scores that have yielded little practical value ([48]), supporting current recommendations to instead target the context in which biased decisions are likely to occur, even before they occur ([2]; [25]). Where bias operates through fast, intuitive judgements, the most reliable strategy is to limit discretion and tie choices to predefined, evidence-based standards. Systemic strategies deliver the most consistent improvements and are typically easier to implement and scale. However, individual strategies add value when they teach the specific cognitive skills a decision might require in realistic simulated settings. For forensic, legal, and other high-stakes systems, where the consequences of error are serious, this combined approach, using systemic design to constrain bias-prone moments and targeted individual training to support improved judgement under pressure, offers the most encouraging route to fairer outcomes.

Beyond the decisions of judges, jurors and other judicial decision-makers, the forensic domain more broadly also warrants attention. Judicial outcomes are often shaped by the evaluations of forensic experts, whose assessments can carry considerable weight in legal reasoning. Emerging evidence indicates that such evaluations are themselves vulnerable to implicit and cognitive biases ([5]), raising the possibility of a cascading effect, in which biases introduced during expert assessment may continue to shape judicial interpretation and reasoning. Our review supports the view that empirical work in this area remains limited, and future research should extend applied testing to forensic expert contexts to examine whether bias-reduction interventions can operate effectively at these earlier stages of decision-making, interrupting potential ‘cascades’ before they reach judicial outcomes. Such work would help clarify how systemic and individual-level strategies can be adapted to strengthen fairness and consistency throughout the justice process.

More broadly, addressing these interconnected sources of bias is essential for translating promising interventions into applied practice and ensuring their impact across professional domains. However, while more applied testing is needed to strengthen the evidence base, especially under realistic constraints, the current evidence base suggests that effective solutions might already exist, but they are yet to be embedded and evaluated in the settings where they are most needed.

## Figures and Tables

**Figure 1 behavsci-15-01592-f001:**
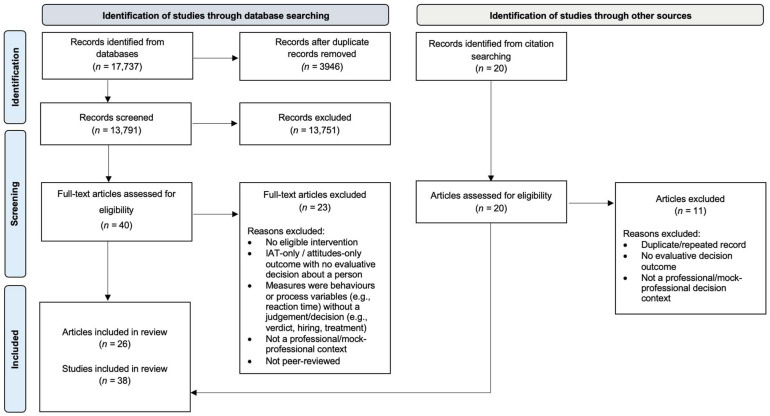
PRISMA flow diagram summarising the literature searching and sifting process.

**Table 1 behavsci-15-01592-t001:** QuADS summary of domain ratings and interpretative summaries.

QuADS Domain	Mean	SD	Range	Interpretive Summary
1. Theoretical or conceptual underpinning to the research	2.97	0.16	2–3	Frameworks were explicitly stated and applied through design and outcomes.
2. Statement of research aim/s	2.97	0.16	2–3	Aims were clearly and explicitly stated in all studies.
3. Clear description of research setting and target population	2.58	0.50	2–3	Settings/populations were described, though contextual detail was often limited.
4. The study design is appropriate to address the stated research aim/s	2.32	0.47	2–3	Designs matched aims but sometimes relied on simplified formats.
5. Appropriate sampling to address the research aim/s	1.71	0.80	1–3	Most lacked strong sampling justification (power, representativeness, recruitment detail).
6. Rationale for choice of data collection tool/s	2.50	0.51	2–3	Most studies provided a rationale for chosen instruments but lacked psychometric validation details.
7. The format and content of the data collection tool are appropriate to address the stated research aim/s	2.58	0.50	2–3	Tools were appropriate and clear; many were author-designed without psychometrics.
8. Description of data collection procedure	2.84	0.37	2–3	Procedures were clearly described; minor omissions occurred.
9. Recruitment data provided	2.05	0.93	1–3	Recruitment/attrition reporting was uneven and often incomplete.
10. Justification for the analytic method selected	2.76	0.63	1–3	Analytic choices were usually well justified and linked to aims.
11. The method of analysis was appropriate to answer the research aim/s	2.84	0.44	1–3	Analyses were generally suitable for the study design and data type.
12. Evidence that the research stakeholders have been considered in the research design or conduct.	0.68	0.74	0–2	Stakeholder engagement was minimal, noted only in a small subset of applied or participatory studies; substantial involvement was absent.
13. Strengths and limitations critically discussed	2.45	0.55	1–3	Most studies provided reflective discussion of limitations, though the depth varied.

## Data Availability

A Searchable Evidence Map (interactive, filterable dataset of study and intervention characteristics and outcomes) is available open-access as Appendix A via OSF https://doi.org/10.17605/OSF.IO/NE4DV. The full Searchable Systematic Map with extracted dataset, coding materials, and working notes is available from the corresponding author on reasonable request. All underlying sources are published studies cited in the article; no new primary data were generated.

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
