# Peer review of "Interventions to Reduce Implicit Bias in High-Stakes Professional Judgements: A Systematic Review"

_behavsci, 2025, doi:10.3390/bs15111592_

Round 1
Reviewer 1 Report
Comments and Suggestions for Authors
The manuscript addresses a timely and highly relevant topic, providing a systematic review that is well structured and supported by a preregistered protocol. The distinction between systemic and individual-level interventions is clearly articulated, and the emphasis on behavioural outcomes rather than attitudinal change represents a notable strength. Nevertheless, several methodological and presentation issues require clarification or improvement.
First, it appears somewhat surprising that the search strategy did not explicitly include terms such as legal or forensic, despite the declared focus of the review on legal and forensic decision-making. The authors are encouraged to specify in detail the exact search strings used for each database, the precise dates of the searches, and the filters applied. The final search date for each database should also be clearly reported to ensure transparency and replicability.
Regarding the appendices, Appendix B is overly verbose and could be made far more effective by providing a concise, structured description of the data extraction methodology. In contrast, Appendix C is not self-sufficient: the full set of criteria and characteristics used in the quality assessment should be presented in detail so that readers can fully understand the process without needing to consult external sources.
The inclusion criteria also warrant clarification. The review refers exclusively to “implicit bias,” but it would be important to explain how this construct was distinguished from other forms of cognitive bias that are not implicit (e.g., heuristics or deliberate distortions) and to justify the decision to restrict the review to implicit bias alone. Similarly, the rationale for limiting the search to English-language studies should be stated explicitly, as this exclusion criterion may have reduced coverage of relevant international literature.
With regard to the presentation of results, while the chapters are adequate, the narrative is fragmented and somewhat difficult to follow. The authors could improve readability by more clearly grouping and synthesising the main findings. A summary table reporting the core characteristics and results of the included studies would greatly enhance clarity, and a separate table presenting the risk-of-bias assessment for each study is strongly recommended to allow readers to evaluate the methodological quality of the evidence base.
Finally, another important point concerns the forensic domain more generally. Judicial decisions are often strongly shaped by the opinions of forensic experts, whose evaluations can weigh heavily in court. Yet, as recent work has shown (Buongiorno et al., 2025 - https://doi.org/10.1016/j.ijlp.2025.102083), forensic psychiatric assessments themselves are not immune to implicit and cognitive biases, including temporal and diagnostic distortions. This raises the possibility of a cascading or “snowball” effect, whereby biases at the expert level may propagate and magnify through subsequent judicial reasoning. It would therefore be valuable for the authors to clarify why such mechanisms have not been considered in the present review, and whether the scope could be expanded to account for these dynamics.
Reviewer 2 Report
Comments and Suggestions for Authors
I was so excited to read this paper – this is really important work, and I think it has been really well executed! A someone who does work in this area, I experience several issues: 1) convincing practitioners that interventions are worth their time and effort even though there are few studies that explicitly test their efficacy and 2) convincing practitioners that participating in rigorous research that demonstrates the efficacy of these interventions will help encourage others to implement the interventions they are nervous to try. This paper shows a need for more work and the weaknesses of the current research while also highlighting the strengths of the existing literature and the types of interventions that show the most promise across disciplines.
In general, I have few comments about the systematic review process itself – it looks to me as though the guidelines they set up in advance were followed, their criteria and approach were sensible, and they followed best practices in reviewing the final set of studies. Well done and thank you for all your hard work on this!
My main comments are mostly suggestions about additional ideas, concepts, or problems you may wish to address. I think that this would strengthen the paper and its impact. At the moment, the take home message falls a little flat and I think that the authors can really say a lot more about what this review suggests about the status quo regarding bias mitigation work. I would classify these suggestions as being on the minor side of major revisions.
- Discussions of implicit bias and cognitive bias: One thing that I have noticed working with practitioners is that they get confused by all the different names for different kinds of biases. How is implicit bias different from contextual, confirmation, and cognitive bias? Are there other names for it? Does it need a different intervention because it is a different kind of bias? I think it would be helpful from the outset to specify that implicit bias is just a name researchers and practitioners use when referring to a specific kind of bias—usually when describing bias arising from demographic factors associated with the target of the decision. Also, there is no definition of cognitive bias (in the general sense) in the paper. I would recommend the one from Spellman, Eldridge, & Bieber (2021: https://doi.org/10.1016/j.fsisyn.2021.100200) at 2.1 (Source of reasoning errors).
- Emphasising the problems that arise from biases occurring outside of people’s awareness: When defining implicit bias and cognitive bias, and when talking about how insufficient awareness training and individual-level/attitudes-focussed interventions are, I think it would be helpful to discuss the unconscious/outside of awareness aspect of bias. That’s what makes bias so hard to prevent (and so interesting to research). It is also one reason why systems-level interventions are more effective than other types of interventions because they don’t rely on individual people to remain aware of and vigilant about their own biases, and because the effort is shared across a system, they are more sustainable too (see Alfaro et al., 2024 https://doi.org/10.1016/j.fsisyn.2024.100569). Related issues that it would be helpful to discuss to hammer home the reason that these interventions are better or worse include bias blind spots (e.g., Kukucka et al., 2017: https://doi.org/10.1016/j.jarmac.2017.09.001), information management procedures (e.g., LSU-E; Dror & Kukucka, 2021: https://doi.org/10.1016/j.fsisyn.2021.100161; Quigley-McBride et al., 2022: https://doi.org/10.1016/j.fsisyn.2022.100216), and the effect of standardising procedures/oversight (Koehler et al., 2023: https://pubmed.ncbi.nlm.nih.gov/37782789/).
- Resource and ethical issues with implementing interventions: In many disciplines, there are issues with resources that prevent a willingness to trial interventions (e.g., public defenders are overworked and haven’t the time to take part in studies that examine their decision-making about clients who are from oppressed demographics, forensic labs are underfunded and already facing significant backlogs and cannot dedicate the hours need to testing a new implementation). The flipside is that they also cannot implement something that has not been thoroughly test if: 1) it could have an impact on their decisions that is unknown in size and direction or 2) they need to answer to what they did in court on cross examination. These two things prevent people in medical, forensic science, and legal fields from trialing bias mitigation interventions much of the time. No one wants to be the first to try an intervention when resources are scarce and there could be implications for the credibility of their work.
- Awareness training as insufficient: I do not think that this point can be made more strongly—so many places are happy to do an awareness training or two but not implement anything systemic or longterm. It is definitely a start and important for people working in the discipline to understand the changes that may be imposed on them (important for buy-in and compliance), but beyond that it is an ineffective safeguard/mitigation strategy.
- Mock-professionals?: I think the weakness of the mock-professional samples should be pointed out more strongly. They are a great starting point, but they do not have the heuristics that an expert sample will have, and that is the entire point of these interventions – use of these samples leaves open the possibility that it would be useless, more effective, or totally different with a group of experts who will have different experience and shortcuts with the same materials.
- Different types of interventions?: It would be helpful for a practitioner audience to provide a table with different types of systemic and individual level interventions and a description of their general features. E.g., information manager/case managers, standardized procedures and oversight, blind proficiency tests of systems and individuals, etc.
- Visuals?: Perhaps some graphs or other visuals of the main findings would be nice and make the data easier to digest?
- Interventions targeting systemic and individual level changes: I suspect that these might be trying to target too many moving parts at once. I haven’t thought it through fully, but there might be something to the fact that changing that many things on that many levels all at once is just too overwhelming.
Reviewer 3 Report
Comments and Suggestions for Authors
The authors present a systematic review of implicit bias interventions with a focus on decision-making contexts in which individual-difference variables (e.g., age, race, gender) could impact important decisions (e.g., high-stakes legal decision making). The goals stated are: (1) identify intervention strategies that have been used, (2) determine the efficacy of these interventions for specific behavioral outcomes (e.g., verdicts or hiring decisions), and (3) evaluate the practicality and transferability of these interventions to specific real-world contexts. This review provides much useful information for researchers exploring implicit bias interventions, as well as suggestions for future research in this area. Below I provide a detailed overview of my review of this work with some recommendations for improvement.
Introduction
The introduction is well written and provides a clear understanding of the issues as well as how this research will add to the field of implicit bias interventions. That said, I do have suggestions for the juror bias section (p. 2). The authors don’t focus on juror decisions in this section; while they do focus on decisions of judges, prosecutors, and defense attorney. I suggest that the authors do the same for jurors and provide a more nuanced discussion of race effects – e.g., Black jurors are biased against White defendants and White jurors against either Black or Hispanic depending on the meta-analysis (see Devine and Caughlin, 2014 and Mitchell et al., 2005 meta analyses). Also, when discussing racial bias for all trial participants (judge/prosecutor/defense attorney/juror) meta-analyses are preferred if available given the inconsistency of findings across individual studies. Finally, is there more recent research on judges’ sentencing decisions as a function of race? I ask because the Mustard (2001) is somewhat dated (almost 25 years old). Other than this the literature cited is timely and appropriate.
Method
The method provides a good amount of information regarding the procedure, but more information is needed in some places. First, could the authors provide a link to their pre-registered review in PROSPERO? It is impossible to assess their adherence to this preregistration without being able to view this registration. Second, the authors discuss PRISMA guidelines, but don’t define what these guidelines are nor a citation for the version of the PRISM guidelines used – see recommendations for doing this here: https://www.prisma-statement.org/citing-prisma-2020 . Third, it would be helpful for the authors to provide the percentage of identified records excluded for each of the indicated exclusion criterium. That is, provide more specific information on how the authors went from 3,946 to 40. This also goes for the reasons for journal articles being dropped from 40 to 23 – reasons are presented but it is not clear the proportion removed for each of the criterial listed. I do appreciate the use of the PRISMA flow chart, but additional information would be helpful. Fourth, a citation is needed for the Quality Assessment Tood for Diverse Studies (QuADS) discussed on p. 7. I appreciate that the authors included the QuADS scoring sheet in Appendix C. Fifth, it would be helpful to have Appendix B introduced prior to the discussion of the four individual domains so that the reader knows when reading each domain that there is more information for each in Appendix B.
Finally, it is not clear why the Supplemental Materials are only available upon request or why the data is available only upon “reasonable request”. This had me thinking about the lack of discussion by the authors of open science in their review of the studies included in the systematic review. Did the authors assess for authors using appropriate open science practices like pre-registration, data availability, and availability of stimuli and measures?
Results
There is a lot of information in the Overview of Study Characteristics section, and it is hard to follow in its current format. A table format would make it easier for the reader to take in this information and have a clear understanding of the main takeaways. I do appreciate the amount of detail provided. Similarly, placing the main findings for the Systemic-Level/Individual-Level/Mixed-Level Interventions in a table would be helpful and would assist the reader in getting a clear understanding of the differences found between these three types of interventions. I understand that this will not be possible for all findings, but suggest the authors consider for the main takeaways because these might get lost in pages of descriptive text.
The following findings may be due to social desirability effects given that category membership is being highlight and the authors should discuss this possibility: “Partitioning candidates by social category encouraged selectors to distribute their selections across categories and outper-formed brief prompts that only stated category information, indicating that the display design, rather than information alone, drove the effect.” Also, the discussion on the Grouping by majority vs. minority was confusing and I am not sure whether I understand what the authors are trying to state. Similarly, more information is needed to understand the studies provided at the end of this paragraph – e.g., what is a “long-hours note” and what was the manipulation in the residency screening and political committee tasks?
More information would be helpful in section 3.4.3 paragraph beginning on line 504 –explain the mediation effect.
As for ecological validity, the authors should discuss (in the Discussion) the roadblocks researchers face when exploring the effects of implicit bias interventions for some types of decisions (e.g., juror and judicial decisions), and some suggestions for overcoming these.
I believe that the authors misstated the cost, facilitation and duration practicality here … “In contrast, the policing programme required about 12 hours of delivery time, 549 access to simulation facilities, and trained instructors, resulting in low cost, facilitation and duration practicality (James et al., 2023).”
Round 2
Reviewer 1 Report
Comments and Suggestions for Authors
The revised manuscript shows effort but does not adequately resolve key methodological and conceptual issues.
The search strategy remains insufficiently transparent and cannot be independently reproduced. The distinction between implicit, cognitive, and cultural bias is still only superficially addressed and not operationalised in the screening process.
The justification for excluding forensic expert assessments weakens the alignment between the stated aims and the actual scope of the review.
Appendices remain incomplete and do not permit verification of quality assessment procedures.
Overall, the revisions improve readability but not methodological rigor.
Reviewer 2 Report
Comments and Suggestions for Authors
The authors chose not to implement any of my suggestions in the manuscript, but explained this decision by noting that their scope was very narrow as was the goal of this paper. While I believe that the manuscript would be improved by emphasizing some aspects of the study/results more and highlighting some of the implications more strongly, thus increasing the impact of the paper, the authors are within their right to stick to a very narrow set of issues if that is their preference.
The addition of the interactive Searchable Evidence Map is a nice touch and does mostly address one of my concerns. I do think that a lot of practitioners might want something more simple (figures/tables) to refer to in order to get a sense of the specific types of interventions examined but, again, if the authors do not feel this aligns with their goals for this paper that is their choice.
Ultimately, now that I know their goal/scope was intentionally narrow, any suggestions I had are really only minor and perhaps not necessary for the journal to accept this paper. I have amended my conclusion to "accept with minor revisions" but the editor may decide that they believe the revisions I suggested are not needed as they seem to be mostly about preferences rather than necessity.
Round 3
Reviewer 1 Report
Comments and Suggestions for Authors
Dear Reviewers,
The revised version addresses all comments, strengthening methodological detail and conceptual coherence. Only minor stylistic edits remain, which will be finalised during proofing.
Kind regards
Author Response
Dear Reviewer 1, thank you once again for such thoughtful and detailed feedback. We truly feel that our systematic review has improved as a result and we are delighted that it will feature in the Special Issue. We hope it will be of interest to many readers.
Best regards, Isabela